# Effects of Adding Inter-Set Static Stretching to Flywheel Resistance Training on Flexibility, Muscular Strength, and Regional Hypertrophy in Young Men

**DOI:** 10.3390/ijerph18073770

**Published:** 2021-04-04

**Authors:** Masatoshi Nakamura, Hirotaka Ikezu, Shigeru Sato, Kaoru Yahata, Ryosuke Kiyono, Riku Yoshida, Kosuke Takeuchi, João Pedro Nunes

**Affiliations:** 1Institute for Human Movement and Medical Sciences, Niigata University of Health and Welfare, 1398 Shimami-cho, Kita-ku, Niigata City, Niigata 950-3198, Japan; hpm19006@nuhw.ac.jp (S.S.); hpm20011@nuhw.ac.jp (K.Y.); hpm19005@nuhw.ac.jp (R.K.); hpm21017@nuhw.ac.jp (R.Y.); 2Central Sports Co., Ltd, 1-21-2 Shinkawa, Chuo-ku, Tokyo 104-8255, Japan; hirotakaikezu0725h@gmail.com; 3Department of Physical Therapy, Faculty of Rehabilitation, Kobe International University, Hyogo 658-0032, Japan; ktakeuchi@kobe-kiu.ac.jp; 4Metabolism, Nutrition, and Exercise Laboratory, Physical Education and Sport Center, Londrina State University, Londrina 86050-070, Brazil; joaonunes.jpn@hotmail.com

**Keywords:** ultrasound, maximal voluntary isometric contraction, concentric strength, eccentric strength, muscle thickness, advanced techniques

## Abstract

Performing static stretching (SS) during resistance training (RT) rest periods is posited to potentiate muscular adaptations, but the literature is scarce on the topic. Thus, the purpose of this study was to investigate the effects of adding inter-set SS to a lower-limb flywheel RT program on joint flexibility, muscular strength, and regional hypertrophy. Sixteen untrained male adults (21 ± 1 y) completed the study, where they performed progressive flywheel bilateral squatting twice a week for 5 weeks. One leg of each participant was randomly allocated to perform SS during the inter-set rest period (RT+SS), while the other leg served as control (RT only). Before and after the intervention, knee flexion range of motion; knee extension isometric, concentric, and eccentric peak torque; 1-repetition maximum; and muscle thickness of the lower-limb muscles were assessed. Following the training period, additional effects were observed for the inter-set SS side on increasing joint flexibility (*p* < 0.05), whereas the average increase in strength measures was 5.3% for the control side, and 10.1% for the inter-set SS side, however, SS intervention induced significantly greater gains only for knee extension isometric strength, but not for dynamic 1-RM, concentric, and eccentric tests. Hamstrings and gluteus maximus did not hypertrophy with training; increases quadriceps muscle thickness depended on the site/portion analyzed, but no significant difference was observed between legs (average: RT = 7.3%, RT+SS = 8.0%). The results indicate that adding inter-set SS to RT may provide large gains in flexibility, slightly benefits for muscular strength (especially for isometric action), but do not impact muscle hypertrophy in untrained young men.

## 1. Introduction

Resistance training (RT) is prescribed to increase muscular strength and hypertrophy in both sports and clinical settings. In addition to traditional RT, several strategies have been investigated in an attempt to potentiate muscular adaptations in shorter time frames [1]. As an example, flywheel RT is a relatively new training method that accentuates the eccentric phase of movement through the inertial force generated by rotation [2,3], and it tends to be more effective for muscle strengthening and hypertrophy [4,5,6], with better results being shown even in the first phases (e.g., 5 weeks) of the training programs [7]. In the same way, adding static stretching (SS) to RT interventions has been used to enhance muscular adaptations [8,9].

The previous study suggested that stretching training using any stretching technique per se could improve the functional test and isotonic muscle strength, not isometric muscle strength [10], and SS training could induce muscle hypertrophy [8]. However, employing long-duration (>60 s) SS exercises could cause a deficit in performance on subsequent RT exercise—the so-called stretching-induced force deficit [11]—thus, the circumstances in which adding SS to RT are determinant to observe negative, equal, or positive responses, as recently noticed by Nunes et al. [8]. Junior et al. [12] showed that carrying out SS immediately before RT sessions did not affect the strength gains, but blunted the hypertrophic responses (compared to a group that performed RT only), and this was attributed to the acute deficits in training performance [8,11,12,13]. Kubo et al. [14] applied SS apart from the RT sessions and showed no difference in muscular responses in relation to the RT-only group. Some authors have suggested that, instead of performing the SS before or distant to the RT session, the SS should be done between the RT sets, i.e., in the inter-set rest period, so this would increase muscle time under tension and neuromuscular, metabolic, and hormonal anabolic responses [9]; factors that are associated with increased strength and hypertrophy [1].

In this sense, two recent studies investigated the effects of performing SS during the RT inter-set rest period [15,16]. One found a significant increase in the muscle thickness of the vastus lateralis, but with no additional effect on hypertrophy of rectus femoris and upper-limb muscles, and strength performance [16], while the other noticed benefits for strength and flexibility [15]. Notably, there is no consensus on the effects of the combination of RT and SS interventions, with potential benefits for adding SS during the RT inter-set rest period [9,15,16], however, further studies are required [8]. In addition, it has been observed that the changes in muscle hypertrophy occur in a heterogeneous way along the muscle length after both RT [17], and SS training programs [18]. Moreover, improvements in strength may be dependent on the type of test used to evaluate such capacity following RT [19,20] and SS [10]. Thus, it remains to be explored whether combining SS and RT can improve muscle hypertrophy in a regional manner and strength in a wider spectrum of tasks.

Therefore, the purpose of this study was to investigate the effect of inter-set SS during flywheel RT on joint flexibility, muscular strength, and regional hypertrophy in young men. It was hypothesized that adding SS to RT would potentiate the adaptations.

## 2. Materials and Methods

### 2.1. Experimental Protocol

Each participant had the dominant (the leg preferred to kick the ball) or the non-dominant leg randomly selected to perform RT with inter-set SS (inter-set SS side), and the other leg was assigned to a control condition (RT only; control side). In this study, eight participants used their dominant side for the inter-set SS side, and the other eight participants used their non-dominant side for the inter-set SS side. This study design was chosen to minimize inter-subjects variability related to personal habits and responsiveness. All participants performed progressive bilateral squat RT using a flywheel training machine (kBOX4 Lite Advanced System, Exxentric AB, Stockholm, Sweden) twice a week for 5 weeks (10 sessions). Knee flexion range of motion (ROM), lower-limb strength (leg press 1-repetition maximum [1RM], and knee extension peak torque [PT] on isometric, concentric, and eccentric tests), and thickness of lower-limb muscles (quadriceps, hamstrings, and gluteus maximus) were measured on the week before (PRE) and on the week after the 5-week training period (POST). At PRE, ultrasonography was performed on days before strength tests; at POST, ultrasonography was performed more than 3-day apart from RT and on days before strength tests. 1-RM and dynamometer tests were performed 72 h apart from each other and the training sessions.

The SS exercises were performed only for the knee extensors (that is why we analyzed only quadriceps muscle thickness in different portions), but hamstrings and gluteus hypertrophy was assessed additionally to verify whether SS added in the inter-set rest period would impair—like when SS is performed just before the RT [12], or not, the responses to all muscles involved on the exercises performed. Moreover, whether these muscles can grow with squat training is a matter of debate in the literature [21].

### 2.2. Participants

Sixteen healthy untrained university male students participated in the study (age: 21.3 ± 1.1 years, stature: 172.5 ± 3.7 cm, body mass: 64.6 ± 8.0 kg). Inclusion criteria were as follows: No regular RT within the past 6 months, no neuromuscular disease, and no history of orthopedic disease for the lower limb. All participants were informed about the procedures and purposes of the study and provided written informed consent. During the experimental period, participants were asked to refrain from any other form of strenuous physical activity than the training performed in the study. This study was conducted under the auspices of the Declaration of Helsinki and approved by the Niigata University of Health and Welfare, Niigata, Japan.

### 2.3. Knee ROM

Each participant assumed a side-lying position on a massage bed with the hip and knee of the non-measurement leg flexed at 90° to prevent movement of the pelvis during the ROM measurements. The investigator brought the measurement leg to full knee flexion with the hip joint in a neutral position. The knee flexion ROM was measured twice using a brass universal goniometer (300 mm), and the average value was considered.

### 2.4. Dynamometry

Participants were seated in a dynamometer chair (Biodex System 3.0, Biodex Medical Systems Inc., Shirley, NY, USA) with the hip flexion angle at 85° and with adjustable Velcro straps fixed over the trunk, pelvis, and thigh of the exercised limb. The knee joint of the exercised limb was aligned with the axis of rotation of the dynamometer. Isometric peak torque (PT-ISO) was measured at two different angles (knee angle of 20° and 70°) using the dynamometer after gravity correction. The participants were instructed to perform maximal contraction for 5 s at each angle two times with a 60-s rest between trials, and the average value was adopted for further analysis. Concentric peak torque (PT-CON) and eccentric peak torque (PT-ECC) were measured at an angular velocity of 60°/s for the ROM of 70° (20°–90° knee angles) for five continuous maximal voluntary concentric contractions in the knee extension direction. The highest value among the five trials was adopted for further analysis. Verbal encouragement was consistently provided during all tests.

### 2.5. One-Repetition Maximum

One-repetition maximum (1-RM) of the unilateral leg press exercise was measured using a horizontal leg-press machine. Participants were instructed to lay on the horizontal leg-press machine with the hip joint flexed 90° and the knee joint flexed to 120°. After several warm-ups, the 1-RM measurement was obtained; the initial weight was selected by each participant’s perceived 1-RM. The weight was increased by 10 kg until the participant could not lift the weight through a full ROM with the proper form, and the 1-RM was identified within five trials. The interval between trials was 2 min, and the investigator verbally encouraged the maximum effort of the participants.

### 2.6. Muscle Thickness

B-mode ultrasonography (LOGIQ e V2; GE Healthcare Japan, Tokyo, Japan) with an 8-MHz linear array probe was used to evaluate the muscle thickness of the quadriceps, hamstrings, and gluteus maximus muscles. For the quadriceps muscle, we measured the distal and proximal thickness of the VL, vastus medialis (VM), RF, and vastus intermedius (VI) muscles in both the lateral and medial regions with the subjects in a supine position with a 0° hip and knee angle of each muscle based on a previous study [17]. In particular, the VI was measured at both the medial and lateral sides because of the possibility of heterogeneous architecture between the medial and lateral regions [22]. The ultrasound measurements were obtained at least 20 min after the participants assumed a supine position. The longitudinal ultrasound images were obtained for the VL, RF, and VI muscles, and the muscle thickness was determined as the mean of the distances between the deep and superficial aponeuroses (for VI, between VI’s superficial aponeurosis and bone) measured at both image ends [17,22]. Regarding the VM muscle, as we could not monitor deep and superficial aponeuroses on the longitudinal image, the transverse ultrasound image was obtained.

The hamstrings and gluteus maximus muscle images were obtained with the subjects in a prone position with the hip and knee angle at 0°. For the hamstrings muscle, we obtained transverse images for each muscle comprising the hamstrings (semitendinosus, semimembranosus, and biceps femoris) midway between the lateral epicondyle and major trochanter of the femur [23]. The average value of these muscles was used as the index of the hamstrings muscles for further analysis. In addition, the gluteus maximus muscle image was obtained at a location 30% proximal to between the posterior superior iliac spine and the greater trochanter [24]. To ensure that the same site was measured before and after the RT period, the ultrasonographic images at POST were taken while referring to the images acquired at PRE.

### 2.7. Training Intervention

All participants performed a bilateral squat progressive RT using an inertial flywheel machine (kBOX4 Lite Advanced System, Exxentric AB, Stockholm, Sweden) twice a week (separated by at least 48 h) for 5 weeks (10 sessions). The training load was increased from a moment inertia of 0.025 kgꞏm^2^ in the first session to 0.100 kgꞏm^2^ in the 10th session. In each session, 3 sets of 10 repetitions of the parallel squat exercise (30 repetitions in total) were performed with a 180-s rest between sets. During the squat exercise, beginning from a squatting position with 90° knee flexion, subjects proceeded to a standing position for 2 s (concentric phase), and then returned to the 90° knee flexion for 2 s (eccentric phase).

During the 180-s rest interval between each set, participants underwent the SS intervention in the SS leg. The SS intervention for the quadriceps muscle was similar to the movement used during the knee flexion ROM measurement. Briefly, the participant assumed a side-lying position on a massage bed with the hip and knee of the control leg flexed at 90° to prevent movement of the pelvis during the SS intervention. The participant brought the inter-set SS side leg to full knee flexion with the hip joint in neutral position at the maximum tolerable stretching intensity. The 30-s SS was repeated twice with a 30-s rest interval, between the first and second set, and the second and third set.

### 2.8. Test–Retest Reliability of the Measurements

The test–retest reliability of the measurement for muscle strength and muscle thickness measurements was determined by the intraclass correlation coefficient (ICC) using six healthy men (age: 23.3 ± 0.7 years; height, 168.0 ± 5.3 cm; and body weight, 60.3 ± 3.4 kg). The ICC of the measurements for PT-ISO, PT-CON, PT-ECC, and 1-RM were 0.91, 0.95, 0.89, and 0.97, respectively. In addition, the ICC of the measurements for the muscle thickness of the distal and proximal VM, VL, RF, lateral VL, and medial VL were 0.993 and 0.996, 0.978 and 0.987, 0.993 and 0.965, 0.987 and 0.961, and 0.955 and 0.97, respectively, and that of the hamstrings and gluteus maximus muscles were 0.956 and 0.955, respectively.

### 2.9. Statistical Analyses

The sample size required for a two-way repeated-measures analysis of variance (ANOVA) (effect size = 0.40 [large], α error = 0.05, and power = 0.80) was calculated using G* power 3.1 software (Heinrich Heine University, Düsseldorf, Germany) based on a previous study, and the required number of participants was greater than 14 for this study. SPSS version 24.0 (IBM Corp., Armonk, NY, USA) was used to conduct the statistical analyses. The normal distribution of the data was confirmed using the Shapiro–Wilk test. Differences in knee ROM, muscle strength, and muscle thickness between the inter-set and control sides were assessed at PRE using paired *t*-tests. For all variables, a two-way repeated ANOVA using two factors [time (PRE vs. POST assessment) and side (inter-set SS vs. control side)] was used to determine the interaction and main effect. If the ANOVA was significant, a post hoc analysis was conducted using a paired *t*-test on each side to determine differences between PRE and POST values. The effect size (ES) was calculated as the difference in the mean value between the PRE and POST values divided by the pooled SD [25]. ES of 0.00–0.19 was considered trivial, 0.20–0.49 was small, 0.50–0.79 was moderate, and ≥0.80 was large [25]. Statistical significance was defined as *p* < 0.05. Descriptive data are reported as mean ± SD.

## 3. Results

All participants completed the RT program with 100% training attendance. Paired *t*-tests showed no significant differences between the intervention and control sides for all variables in the PRE value.

### 3.1. Changes in ROM, PT-ISO, PT-CON, PT-ECC, and 1-RM

Table 1 shows the changes in ROM, PT-ISO, PT-CON, PT-ECC, and 1-RM before and after 5 weeks of RT in both the inter-set SS and control sides. The ANOVA showed a significant interaction effect for knee flexion ROM (F = 4.85, *p* = 0.044, η_p_^2^ = 0.244), and the post hoc test showed a significant difference between the PRE and POST values on the inter-set SS side (*p* < 0.01, d = 1.63), but not on the control side (*p* = 0.81, d = 0.10). The change in knee flexion ROM between PRE and POST measurements on the inter-set SS side was significantly higher than that on the control side (*p* = 0.04). Similarly, the ANOVA showed a significant interaction effect for PT-ISO (F = 10.6, *p* < 0.01, η_p_^2^ = 0.415), and the post hoc test showed significant increases between PRE and POST values on both the inter-set SS and control sides (*p* < 0.01, d = 0.70, and *p* = 0.01, d = 0.53, respectively). The change between PRE and POST measurement on the inter-set SS side was significantly higher than that on the control side (*p* < 0.01). Although there was no significant interaction effect of PT-CON (F = 0.33, *p* = 0.575, η_p_^2^ = 0.021), there was a significant main effect of time (F = 13.3, *p* < 0.01, η_p_^2^ = 0.469). The post hoc test showed significant increases between PRE and POST values on both inter-set SS and control sides (*p* = 0.01, d = 0.56, and *p* < 0.01, d = 0.52, respectively). For PT-ECC, there was no main effect of time or interaction effect (F = 1.4, *p* = 0.251, η_p_^2^ = 0.087 and F < 0.01, *p* = 0.998, η_p_^2^ < 0.01, respectively). Regarding the leg press 1-RM, there was no significant interaction effect (F = 1.36, *p* = 0.262, η_p_^2^ = 0.083), but there was a significant main effect of time (F = 11.3, *p* < 0.01, η_p_^2^ = 0.43). The post hoc test showed significant increases between PRE and POST values on both the inter-set SS and control sides (*p* = 0.01, d = 0.60, and *p* = 0.03, d = 0.46, respectively).

### 3.2. Changes in Muscle Thickness of Quadriceps, Hamstrings, and Gluteus Maximus Muscles

The results of muscle thickness are shown in Table 2. The ANOVA revealed no significant interaction effects for all muscle thickness measures of the quadriceps (VL distal: F = 1.29, *p* = 0.274, η_p_^2^ = 0.079, VL proximal: F = 0.06, *p* = 0.810, η_p_^2^ < 0.01; VM distal: F = 2.79, *p* = 0.12, η_p_^2^ = 0.157, VM proximal: F = 1.52, *p* = 0.24, η_p_^2^ = 0.09; RF distal: F = 0.31, *p* = 0.59, η_p_^2^ = 0.02, RF proximal: F = 1.35, *p* = 0.26, η_p_^2^ = 0.08, lateral VI distal: F < 0.01, *p* = 0.93, η_p_^2^ < 0.01, lateral VI proximal: F = 4.4, *p* = 0.06, η_p_^2^ = 0.23, medial VI distal: F = 0.49, *p* = 0.50, η_p_^2^ = 0.03, medial VI proximal: F = 0.03, *p* = 0.86, η_p_^2^ < 0.01), hamstrings (F = 0.06, *p* = 0.80, η_p_^2^ < 0.01), and gluteus maximus muscles (F = 0.14, *p* = 0.71, η_p_^2^ = 0.01). The main effects of time were found for the muscle thickness of the VM in the proximal and distal regions (F = 21.8, *p* < 0.01, η_p_^2^ = 0.59, and F = 13.9, *p* < 0.01, η_p_^2^ = 0.48, respectively), VL in the proximal region (F = 52.4, *p* < 0.01, η_p_^2^ = 0.78), RF in both the proximal and distal regions (F = 16.5, *p* < 0.01, η_p_^2^ = 0.52, and F = 17.6, *p* < 0.01, η_p_^2^ = 0.54, respectively), and medial portion of the VI in both the proximal and distal regions (F = 20.9, *p* < 0.01, η_p_^2^ = 0.58, and F = 6.3, *p* = 0.02, η_p_^2^ = 0.29, respectively). The post hoc test showed significant increases in the muscle thickness of the VM in the proximal and distal regions in both the inter-set SS and control sides (distal region: Inter-set SS side: *p* < 0.01, d = 0.67, and control side: *p* < 0.01, d = 0.47, proximal region: Inter-set SS side: *p* < 0.01, d = 0.60, and control side: *p* = 0.03, d = 0.39), and VL in the proximal region (inter-set SS side: *p* < 0.01, d = 0.96, and control side: *p* < 0.01, d = 0.94), RF in both the proximal and distal regions (distal region: Inter-set SS side: *p* < 0.01, d = 0.78, and control side: *p* < 0.01, d = 0.71, proximal region: Inter-set SS side: *p* < 0.01, d = 0.59, and control side: *p* = 0.09, d = 0.34), and medial portion of the VI in both the proximal and distal regions (distal region: Inter-set SS side: *p* < 0.01, d = 0.79, and control side: *p* < 0.01, d = 0.93, proximal region: Inter-set SS side: *p* = 0.04, d = 0.46, and control side: *p* = 0.05, d = 0.61).

## 4. Discussion

In the present study, we investigated the effects of adding inter-set SS during flywheel RT on changes in lower-limb flexibility, strength, and hypertrophy and, especially, the regional hypertrophy of the four heads of the quadriceps. Our results showed that inter-set SS may potentiate flexibility as well as isometric knee strength, whereas there were no significant additional increments in dynamic strength (PT-CON, PT-ECC, and 1-RM) and muscle hypertrophy in the quadriceps muscles. Although previous studies indicated that inter-set SS could achieve additional results [15,16], this is the first paper to investigate the effect of inter-set SS intervention on muscle strength in different contraction modes and muscle hypertrophy in different regions.

Regarding ROM, our results revealed that a significant increase in knee flexion ROM was found only on the inter-set SS intervention side, which was expected due to the nature of the SS training. Many previous studies have shown increases in ROM after chronic SS training [26,27], consistent with our results. The actual mechanism for the increase in knee flexion ROM has been unclear, although it is often attributed to increases in stretch tolerance or reductions in muscle stiffness. Nonetheless, a previous study proposed that, within short duration intervention periods (<8 weeks), the increases in ROM could be a result of changes in stretch tolerance, rather than changes in passive muscle stiffness [27].

Regarding the changes in strength, our results showed increments over time in both legs for 1-RM, PT-ISO, and PT-CON, but not for PT-ECC. The average increase in strength capacity (considering all the tests done) was 5.3% for the control side, and 10.1% for the inter-set SS side, however, SS intervention induced significantly greater gains only in knee extension PT-ISO, with no additional effect for leg press 1-RM, PT-CON, and PT-ECC. It is possible that a longer intervention length could induce significant advantages for adding SS to RT, but further studies are needed to test such a hypothesis. The previous studies that investigated the effects of inter-set SS on muscle strength showed that the strength increase for inter-set SS conditions was limited or absent. Some authors recently suggested that inter-set SS could increase the total time under tension for the muscle and increase the neuromechanical and metabolic stimuli, which is important for muscle strengthening [9]. Therefore, we hypothesized that the inter-set SS could enhance muscle strength gains, but our hypothesis was only partially confirmed. This discrepancy could be due to the difference in muscle contraction type during SS and RT, and the additional increment effect could differ depending on the different contraction styles (dynamic vs. isometric contraction).

In the present study, we demonstrated that a short-term flywheel squat RT could induce muscle hypertrophy in the quadriceps muscles, but not in the hamstrings and gluteus maximus, and that inter-set SS could not induce an additional increase or regional difference. Similar to the increase in muscle strength, Mohamad et al. [9] suggested that the duration of muscle tension brought by the inter-set SS would be important in maximizing the muscle hypertrophic response [9]. In fact, Evangelista et al. [15,16] showed that an 8-week inter-set SS during traditional RT could enhance the muscle hypertrophy of VL. However, we did not observe additional effects with SS and speculate that this might have occurred because flywheel RT tends to produce large responses than traditional RT [7], so that a ceiling effect was observed with the condition without SS. Moreover, our previous studies [28,29] and review [8] showed that passive low-intensity SS interventions often do not cause significant increases in muscle thickness. Future studies should consider adding loaded SS to RT [8].

There were some limitations in the present study. The training period was only 5 weeks; longer duration studies are needed. Moreover, dietary intake and daily physical activity levels were not assessed, and whether these factors could exert some influence on the adaptations remains uncertain. Finally, this experiment was performed in untrained young adult men, and results cannot be generalized to other populations of different sex, age, or training status.

## 5. Conclusions

In conclusion, our results indicate that adding inter-set SS to RT may provide large gains in flexibility and slight benefits for muscular strength (especially for isometric action), but do not impact muscle hypertrophy in untrained young men. Inter-set SS intervention may increase joint flexibility and enhance the increase in isometric strength. As the inter-set SS intervention does not require special equipment or additional time, coaches and practitioners can use this strategy in training and rehabilitation settings with some positive effects on muscular outcomes.

## Figures and Tables

**Table 1 ijerph-18-03770-t001:** Results following training period on knee range of motion (ROM), leg-press one-repetition maximum (1-RM), knee extension isometric peak torque (PT-ISO), concentric peak torque (PT-CON), and eccentric peak torque (PT-ECC) in young men (n = 16).

Variables	Condition	PRE	POST	Effect Size	%Diff
Knee flexion	Control	148.2 ± 3.9	148.8 ± 6.5	0.10	0.5
ROM (°)	Inter-set SS	146.3 ± 4.4	152.9 ± 3.8 **	1.63	4.5
Leg press	Control	73.1 ± 9.0	77.8± 11.6 *	0.46	6.4
1-RM (kg)	Inter-set SS	71.9 ± 10.9	79.4 ± 14.0 *	0.60	10.4
Knee extension	Control	138.9 ± 24.3	150.7 ± 20.0 *	0.53	8.5
PT-ISO (Nm)	Inter-set SS	137.7 ± 31.0	158.9 ± 29.3 **	0.70	15.4
Knee extension	Control	166.9 ± 32.7	183.3 ± 30.2 **	0.52	9.8
PT-CON (Nm)	Inter-set SS	173.8 ± 38.6	193.3 ± 31.0 *	0.56	11.2
Knee extension	Control	209.9 ± 43.3	202.4 ± 21.5	−0.23	−3.6
PT-ECC (Nm)	Inter-set SS	220.3 ± 54.3	227.7 ± 48.9	0.14	3.4

Notes. PRE = before training program; POST = after training program. * *p* < 0.05 vs. PRE; ** *p* < 0.01 vs. PRE.

**Table 2 ijerph-18-03770-t002:** Results following training period on muscle thickness (mm) of quadriceps, hamstrings, and gluteus maximus muscles in young men (n = 16).

Variables		Condition	PRE	POST	Effect Size	%Diff
VL	Distal	Control	22.1 ± 4.1	23.6 ± 4.7	0.35	6.7
		Inter-set SS	20.2 ± 3.3	21.4 ± 3.3	0.38	5.9
	Proximal	Control	23.1 ± 3.4	26.0 ± 2.8 **	0.96	12.6
		Inter-set SS	23.0 ± 3.0	26.1 ± 3.7 **	0.94	13.5
VM	Distal	Control	25.8 ± 4.6	27.7 ± 3.7 **	0.47	7.4
		Inter-set SS	25.6 ± 4.9	28.3 ± 4.0 **	0.67	10.5
	Proximal	Control	31.8 ± 4.7	33.7 ± 3.8 *	0.39	6.0
		Inter-set SS	31.2 ± 4.5	33.7 ± 3.8 **	0.60	8.0
RF	Distal	Control	19.4 ± 2.8	21.2 ± 2.1 **	0.71	9.3
		Inter-set SS	19.6 ± 3.4	21.7 ± 1.8 **	0.78	10.7
	Proximal	Control	23.5 ± 2.5	24.2 ± 2.0	0.34	3.0
		Inter-set SS	23.6 ± 2.4	24.9 ± 2.1 **	0.59	5.5
VI lateral	Distal	Control	15.7 ± 3.3	16.5 ± 4.2	0.21	5.1
		Inter-set SS	17.4 ± 4.6	18.2 ± 5.0	0.15	4.6
	Proximal	Control	18.9 ± 5.3	19.0 ± 4.7	0.01	0.5
		Inter-set SS	19.6 ± 5.8	17.9 ± 4.0	−0.34	−8.7
VI medial	Distal	Control	16.6 ± 2.3	19.0 ± 2.7 **	0.93	14.5
		Inter-set SS	16.0 ± 2.6	17.8 ± 2.1 **	0.79	11.3
	Proximal	Control	21.8 ± 3.5	23.5 ± 3.8 *	0.46	7.8
		Inter-set SS	21.2 ± 3.4	23.0 ± 2.6 *	0.61	8.5
Hamstrings		Control	27.0 ± 3.2	27.3 ± 2.8	0.10	1.1
		Inter-set SS	26.4 ± 3.2	26.9 ± 3.2	0.16	1.9
Gluteus maximus		Control	22.6 ± 4.0	22.3 ± 6.6	−0.06	−1.3
		Inter-set SS	23.1 ± 4.9	22.3 ± 3.8	−0.19	−3.5

Notes. PRE = before training program; POST = after training program. VL = vastus lateralis; VM = vastus medialis; RF = rectus femoris; VI = vastus intermedius. * *p* < 0.05 vs. PRE; ** *p* < 0.01 vs. PRE.

## Data Availability

All data generated or analyzed during this study are included in this published article.

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
