# Peer review of "Effects of Adding Inter-Set Static Stretching to Flywheel Resistance Training on Flexibility, Muscular Strength, and Regional Hypertrophy in Young Men"

_ijerph, 2021, doi:10.3390/ijerph18073770_

Round 1
Reviewer 1 Report
Pag. 2 refer that
“The SS training per se can induce gains in strength [10]”
The author does not consider that SS training induce strength, he consider positive effect in muscle performance, and also does not say that SS training but flexibility training (FT) that is a wider range of methodologies and not the specific Static Stretch.
He consider that “The improvements were observed only in functional tests and isotonic contractions, isometric contractions were not affected by FT. Therefore, FT might have an influence on dynamic MP. However, more studies are necessary to confirm whether FT can positively affect MP”
Also in Pag. 2 refer thaT two studies that get different results, but in the explanation the author puts both studies in the same comment!
“In this sense, two recent studies investigated the effects of performing SS during the RT inter-set rest period [15, 16]. One found a significant increase in the muscle thickness of the vastus lateralis, but with no additional effect on hypertrophy of rectus femoris and upper-limb muscles, and strength performance [15, 16], while the other noticed benefits for strength and flexibility [15].
Author Response
Reviewer #1
We thank the reviewers for the constructive comments and suggestions on our manuscript titled “Effects of Adding Inter-Set Static Stretching to Flywheel Resistance Training on Flexibility, Muscular Strength, and Regional Hypertrophy in Young Men” We used a red font for our responses to the reviewers’ comments and for quoted texts from the original manuscript, with revisions underlined.
Pag. 2 refer that “The SS training per se can induce gains in strength [10]”
The author does not consider that SS training induce strength, he consider positive effect in muscle performance, and also does not say that SS training but flexibility training (FT) that is a wider range of methodologies and not the specific Static Stretch.
He consider that “The improvements were observed only in functional tests and isotonic contractions, isometric contractions were not affected by FT. Therefore, FT might have an influence on dynamic MP. However, more studies are necessary to confirm whether FT can positively affect MP”
Response: Thank you very much for your indication. As you pointed out, the previous study by Medeiros et al., (2017) suggested that flexibility training using any stretching technique could improve functional test and isotonic contraction, not in isometric contraction strength. Therefore, we revised the sentence accordingly as below:
Change: Lines 45-47 in revised manuscript
The previous study suggested that stretching training using any stretching technique per se could improve the functional test and isotonic muscle strength, not isometric muscle strength [10], and SS training could induce in muscle hypertrophy [8].
Also in Pag. 2 refer that two studies that get different results, but in the explanation the author puts both studies in the same comment!
“In this sense, two recent studies investigated the effects of performing SS during the RT inter-set rest period [15, 16]. One found a significant increase in the muscle thickness of the vastus lateralis, but with no additional effect on hypertrophy of rectus femoris and upper-limb muscles, and strength performance [15, 16], while the other noticed benefits for strength and flexibility [15].
Response: Thank you very much for finding our mistake, and we revised the sentence accordingly in Lines 60-62.
Reviewer 2 Report
Very interesting, beneficial, and seriously processed manuscript. It's just a pity that the number of probands is relatively low.
I have only the following comments or suggestions for the whole work:
- The ROM has been evaluated, it is necessary to state which type of goniometer was used
- For probands, one lower limb was randomly selected, it would be appropriate to state "how many dominant and how many non-dominant limbs were" and whether dominance or non-dominance had / could affect the results
- The limitations state that the physical activities of the probands were not taken into account, if it would be possible to indicate at least the occupations of the probands or whether their occupations differed
- 5 weeks were used for the intervention, I miss the reason why just 5 weeks and why the frequency of exercise twice a week
Author Response
Reviewer #2
Very interesting, beneficial, and seriously processed manuscript. It's just a pity that the number of probands is relatively low.
I have only the following comments or suggestions for the whole work:
We thank the reviewers for the constructive comments and suggestions on our manuscript titled “Effects of Adding Inter-Set Static Stretching to Flywheel Resistance Training on Flexibility, Muscular Strength, and Regional Hypertrophy in Young Men” We used a red font for our responses to the reviewers’ comments and for quoted texts from the original manuscript, with revisions underlined.
-The ROM has been evaluated, it is necessary to state which type of goniometer was used
Response: Thank you very much for your indication. We used the universal goniometer, and we added the information as below:
Changes: lines 108-109
The knee flexion ROM was measured twice using a brass universal goniometer (300 mm), and the average value was considered.
-For probands, one lower limb was randomly selected, it would be appropriate to state "how many dominant and how many non-dominant limbs were" and whether dominance or non-dominance had / could affect the results.
Response: In this study, the inter-SS side was allocated for dominant side in eight participants and non-dominant side in eight participants, respectively, and we added the following description. In addition, a previous study (Carvalho et al., JSMPF. 2020) showed that there were no significant changes in muscle strength and muscle thickness between dominant and non-dominant limbs after resistance training. Therefore, we believe that dominant or non-dominant limbs for inter-set SS side could not affect the results of current study.
Changes: lines 77-79
In this study, eight participants used their dominant side for the inter-set SS side, and the other eight participants used their non-dominant side for the inter-set SS side.
-The limitations state that the physical activities of the probands were not taken into account, if it would be possible to indicate at least the occupations of the probands or whether their occupations differed
Response: In this study, all participants are healthy young sedentary university students, and during the experimental period, participants were asked to refrain from any other form of strenuous physical activity than the training performed in the study. We added the description to clarify this point.
Changes: lines 96-101
Sixteen healthy untrained university male students participated in the study (age: 21.3 ± 1.1 years, stature: 172.5 ± 3.7 cm, body mass: 64.6 ± 8.0 kg). Inclusion criteria were as follows: no regular RT within the past 6 months, no neuromuscular disease, and no history of orthopedic disease for the lower limb. All participants were informed about the procedures and purposes of the study and provided written informed consent. During the experimental period, participants were asked to refrain from any other form of strenuous physical activity than the training performed in the study.
- 5 weeks were used for the intervention, I miss the reason why just 5 weeks and why the frequency of exercise twice a week
Response: The previous studies [4-6] showed that more than 5 weeks flywheel RT could induce the increases in muscle strength and muscle mass, and we adopted the 5 weeks intervention periods (please see in lines 39-43).